# Constructing a Shariah Document Screening Prototype Based on Serverless Architecture

**Marhanum Che Mohd Salleh** [1,*], **Rizal Mohd Nor** [2] , **Faizal Yusof** [3] **and Md Amiruzzaman** [4]

1 Department of Finance, International Islamic University Malaysia, Kuala Lumpur 53100, Malaysia
2 Department of Computer Science, International Islamic University Malaysia, Kuala Lumpur 53100, Malaysia
3 Faculty of Resilience, Rabdan Academy, 65 Al Inshirah Street, Al Sa'adah,
  Abu Dhabi 22401, United Arab Emirates
4 Department of Computer Science, West Chester University, West Chester, PA 19383, USA
* Correspondence: marhanum@iium.edu.my

**Abstract:** The aim of this research is to discuss the groundwork of building an Islamic Banking Document Screening Prototype based on a serverless architecture framework. This research first forms an algorithm for document matching based Vector Space Model (VCM) and adopts Levenshtein Distance for similarity setting. Product proposals will become a query, and policy documents by the central bank will be a corpus or database for document matching. Both the query and corpus went through preprocessing stage prior to similarity analysis. One set of queries with two sets of corpora is tested in this research to compare similarity values. Finally, a prototype of Shariah Document Screening is built based on a serverless architecture framework and ReactJS interface. This research is the first attempt to introduce a Shariah document screening prototype based on a serverless architecture technology that would be useful to the Islamic financial industry towards achieving a Shariah-compliant business. Given the development of Fintech, the output of this research study would be a complement to the existing Fintech applications, which focus on ensuring the Islamic nature of the businesses.

**Keywords:** Islamic banking; document screening; Levenshtein distance; serverless architecture





## 1. Background of Research

Digitalization is transforming how people interact and conduct daily business. The advancements in banking technology have influenced the future of financial services around the world. The establishment of digital banking is now in its growing stage where the way financial services are offered to all is positively accepted, especially by our new generation of digitally savvy millennials and Gen Zers. Through technology, the business is leaning more towards business-to-customer (BTC), where the financial services are directly reached to the customer at a fingertip and within seconds. The usage of artificial intelligence (AI) has complimented the roles of humans as financial advisors, and through blockchain technology, the whole financial system becomes efficient to tie all segments of banking businesses. Apart from the finance industry, Machine learning (ML), being a vast area within artificial intelligence (AI), has revolutionized many industries and fields. Meta-heuristic optimization algorithm based on a hybrid of the Sine Cosine Algorithm (SCA) and the Grey Wolf Optimizer (GWO) can be used to train the Multilayer Perceptron (MLP) neural network [1].

Despite all the advantages that technology has brought to the financial market, there are a few segments that still need attention to be digitalized, especially the Islamic banking business, which requires attention on Shariah compliancy in all aspects. It is observed that there is a limitation of technology usage in product structuring, mainly on documentation, which is conducted manually by bank officers. Compared to conventional products, Islamic banking products require more documents based on the nature of the contracts adopted.

It is the duty of Shariah officers to ensure that all documents comply with the Shariah and the central bank policy in the process of structuring new products. This duty would require much effort, time, and energy to verify that the product will not have any element of Shariah non-compliance. Hence, the advancement of Natural Language Processing (NLP) technology would be necessary and significant to assist this process and would reduce the burden of the Shariah officers. The existence of serverless technology would ensure the process of software development for Shariah screening is shortened. The development team can release applications, analyze user feedback, and iterate improvements faster.

Obviously, there are a limited number of Robo Advisors offered in the financial market, and the focus is merely on digital investment management services. Based on observation, there are five Robo Advisors currently in the Malaysian financial market, which are MyTheo, StashAway, BEST, Wahed Invest, and Raiz. Unfortunately, none of these Robo Advisors have aided the Islamic banks regarding the Shariah-compliant status of the products during the structuring process. Based on the suggestion by [2], the involvement of Robo advisory will ease the product structuring process by evaluating the sources of Shariah to provide necessary information for the physical Shariah advisor to accomplish the ruling accordingly. This effort may be started with the development of an NLP algorithm for document matching or similarity screening.

Moreover, there are several Shariah non-compliant cases among Islamic banks in Malaysia where the root cause is documentation issues [3]. As reported, most of the cases are related to the contract documentation, calculation of Ta'widh/Ibra', and term of conditions which contradict the nature of the contract (sample of cases: CIMB Islamic Bank Bhd v LCL Corp Bhd & Anor [2012] 3 MLJ 869 and one Court of Appeal case of Pan Northern Air Services Sdn Bhd v Maybank Islamic Bhd and another appeal [2021] 3 MLJ 408). The existence of an efficient platform (a mechanism) to retrieve relevant documents efficiently would assist the Shariah officer as well as the Shariah Committee in conducting earlier screening of the Shariah status of the documents/products proposed.

This paper is structured in the following manner: it starts with a research background and is followed by a discussion of the past literature, mainly on financial technology adoption in the banking industry, updates on the NLP literature, and serverless architecture. This is followed by research methodology and the suggestion of a Shariah document screening prototype. It ends with a discussion and suggestions for future research.

## 2. Past Literature

### 2.1. Adoption of Financial Technology (Fintech) in the Financial Industry

The emergence of the Islamic financial industry since the 1960s has brought the practice of a dual banking system to the world. The system has been recognized not only in Muslim countries but also by non-Muslim countries. This recognition is something that Muslims must be proud of as the world has respected Islamic law (Shariah) as the main law for Islamic financial businesses. This is because the aim of Shariah is to safeguard all aspects of human beings, and this is the missing part of the conventional financial system. The objective of the Shariah (Maqasid Shariah), which is comprised of five elements (religion/belief, lineage, wealth, intellect, and life), became the main aspect whenever Islamic financial products were offered in the market.

Accordingly, with the strong support from the authority and Shariah advisory committees both at the local and international level, such as the Shariah Advisory Council of the Central Bank of Malaysia and the International Islamic Fiqh Academy, which is comprised of 57 member states of the Organization of Islamic Cooperation (OIC), any issues which arise pertaining to the industry is discussed and solved at various levels. Now, the industry has been developed, remains competitive with its conventional counterparts, and is evolving with financial technology. As an update on the Malaysian Islamic financial industry, the central bank granted five digital banks licenses recently on April 2022, which are;

Licensed under the Financial Services Act 2013 (FSA):

i. a consortium of Boost Holdings Sdn. Bhd. (Kuala Lumpur, Malaysia) and RHB Bank Berhad;

ii. a consortium led by GXS Bank Pte. Ltd. (Singapore) and Kuok Brothers Sdn. Bhd (Kuala Lumpur, Malaysia); and

iii. a consortium led by Sea Limited (Singapore) and YTL Digital Capital Sdn Bhd (Kuala Lumpur, Malaysia).

Licensed under the Islamic Financial Services Act 2013 (IFSA):

i. a consortium of AEON Financial Service Co., Ltd. (Tokyo, Japan), AEON Credit Service (Hong Kong, China) (M) Berhad and MoneyLion Inc. (New York, NY, USA); and

ii. a consortium led by KAF Investment Bank Sdn. Bhd.

Three of the five consortiums are majority-owned by Malaysians, namely Boost Holdings and RHB Bank Berhad, Sea Limited, and YTL Digital Capital Sdn. Bhd. and KAF Investment Bank Sdn. Bhd.

## 2.2. Fintech and Banking Business

The current fintech environment has opened unprecedented opportunities for banking businesses and their customers. It is undeniable that financial services have become faster and easy to use where money can be transferred within a few seconds, and financing products can be approved within an hour. Technology has replaced human roles in providing financial services. Scholars have agreed that fintech somehow has indirectly brought negative effects to the traditional banking system [4,5]. Fintech has brought positive effects to the banking system, as reported by researchers, such as a digital calculator for bank charges using information asymmetry [6], detection of anomalies in data streaming [7], digital signature for bank customers using a functional symmetry approach [8], and others.

In Malaysia, the government, through the central bank, has supported the emergence of technology in the Malaysian financial system. Among the initiative of the bank is to capitalize on digitalization which is stated in Malaysia Financial Sector Blueprint 2022–2026. The bank has provided a few technology infrastructures as a backbone of the digital economy, which includes real-time payment systems and a few guidelines on fintech. After the successful establishment of digital banks, the central bank issued a discussion paper on the Licensing Framework for Digital Insurers and Takaful operators in February 2022. Other standards or guidelines that were set by the bank are the policy on Risk Management in Technology (2020), the Financial Technology Regulatory Sandbox Framework (2016), and the Minimum Guideline on the Provision of Internet Banking Services by Licenced Baking Institutions (2014).

Overall, the technology has also opened the door to new businesses related to financial services (fintech business), which has become a challenge to the existing financial institutions. However, it is observed that the Malaysian financial market is healthy enough to welcome newcomers where they have complimented each other and even become strategic partners. In this context, the Malaysian government has been supportive, and among the initiatives given to the small and medium enterprises fintech companies are tax angel incentives (granted to angel investors in fintech start-ups), income tax exemption by Malaysian Industrial Development Authority (MIDA), partial corporate tax exemption for entities in the Malaysian Digital Hub under MDEC, Malaysia Tech Entrepreneur Program under MDEC, Multimedia Super Corridor (MSC) Malaysia status recognition for ICT, and others.

In the Islamic finance industry [9], we have suggested the usage of AI in NLP based on the Islamic FinTech Model that combines Zakat and Qardh-Al-Hasan (benevolent loan) to minimize the negative impact of COVID-19 on individuals and SMEs. The authors believed that Islamic finance has big potential to face any kind of situation/pandemic, especially the combination of Zakat and Qardh-Al-Hasan. In addition, research conducted by [10] and others [11,12] have explored the potential of Fintech, which includes AI, smart contract, blockchain, and others in the Islamic banking and finance industry in various Asian countries and their findings indicate that Fintech would benefit the industry greatly to be at par with the conventional counterparts. Hence, a lot of effort is needed to implement

Fintech in the Islamic financial industry so that it would have a significant impact on society and be practiced with the Shariah spirit.

Figure 1 presents the increasing trend in the literature on NLP based on the Scopus database in all areas of research, including social science, arts, engineering, computer sciences, business, management, economics, and others. Since 2016, the number of literature studies continued to increase until 2021 (2045 articles) and is expected to increase further until the end of this year. Since 1976, a total of ten thousand articles have been published in the Scopus database, and the author that has contributed a lot in this domain is Liu, H, with 77 articles. It is followed by Xu, H and Friedman, C. Please refer to Figure 2.

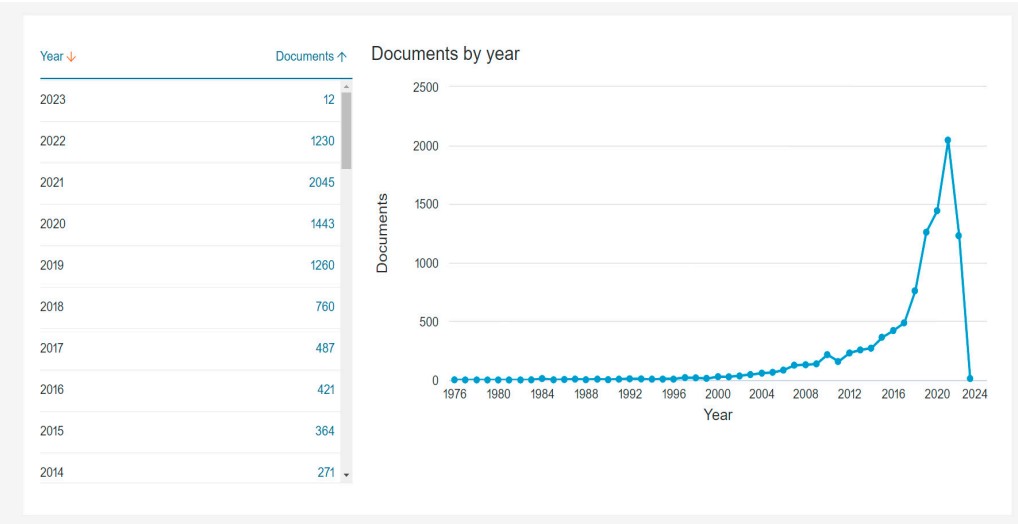

**Figure 1.** Number of literature studies on NLP by year.

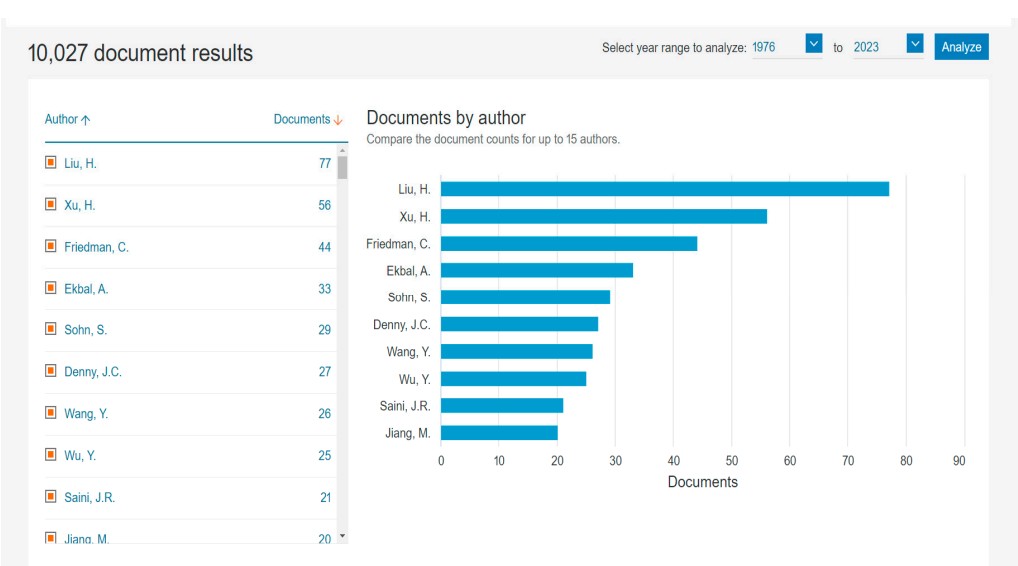

**Figure 2.** Total number of NLP literature studies by author.

Accordingly, as presented in Figure 3 below, the United States has been dominant in discussing NLP (2642 articles), followed by China (1542 articles), India (1356 articles), and the United Kingdom (573 articles). It is undeniable that these countries have been at the forefront of technology adoption. It is also not a surprise that NLP is mainly conducted by researchers from computer science (37.3%) and engineering (14.3%). Other areas that have adopted NLP are medicine, social science, arts, physics, business, management, and accounting. Please refer to Figure 4

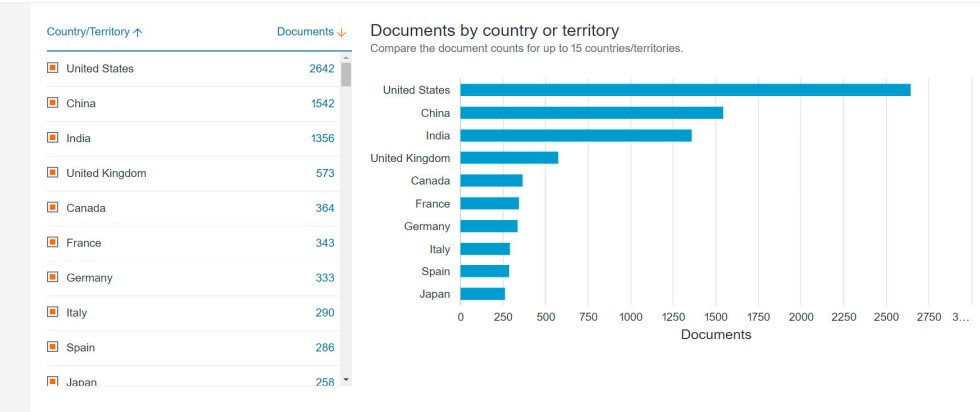

**Figure 3.** Countries that have contributed to the NLP literature.

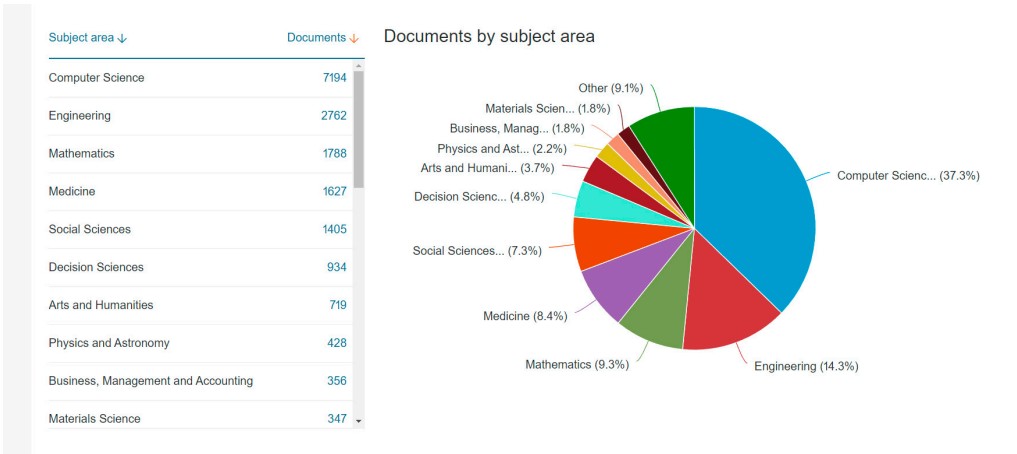

**Figure 4.** The literature on NLP by subject area.

In addition, out of ten thousand literature studies on Natural Language Processing (NLP) from various areas, this research has limited the search of the subject area to only social sciences, business, management, and accounting. As in Figure 4, there are 1712 articles found in these areas. Bhattacharyya, P., Ekbal, A., and Korhonen, A. have contributed equally to the literature.

This research further defined the search for the economics, finance, business, management, and accounting subject areas (please refer to Figure 5). The results indicate that there are 178 articles with an increasing trend, which is expected to increase until 2022. Please refer to Figure 6. The main contributors are Coffas, Delcea, and Melumad, and each of them wrote three articles on NLP between the year 2019–2020 (please refer to Figure 7). Most authors have adopted NLP in social media research, either in terms of sentiment analysis based on image evaluation, customers' product opinions, social media users' emotions, or on consumers' vocabularies. The most cited article was by Melumad (2019) on the effect of social media content on the usage of smartphones. Based on the literature findings, there are still limited literature studies that have utilized NLP for banking and finance.

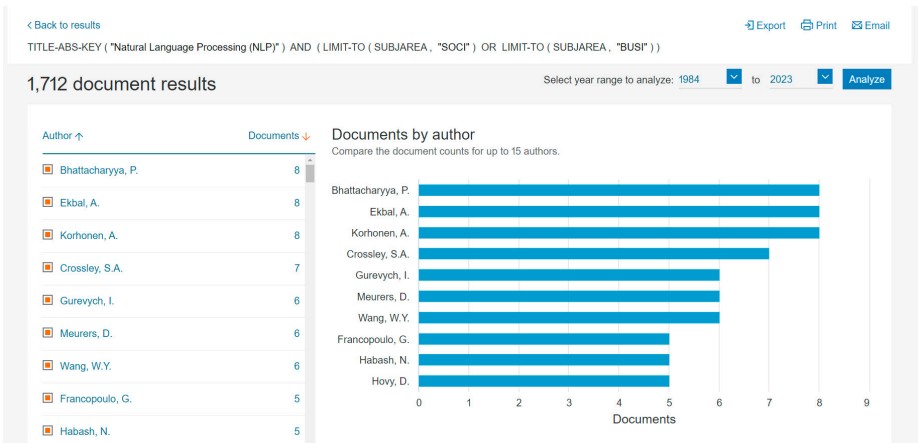

**Figure 5.** NLP literature studies in social sciences, business, management, and accounting area.

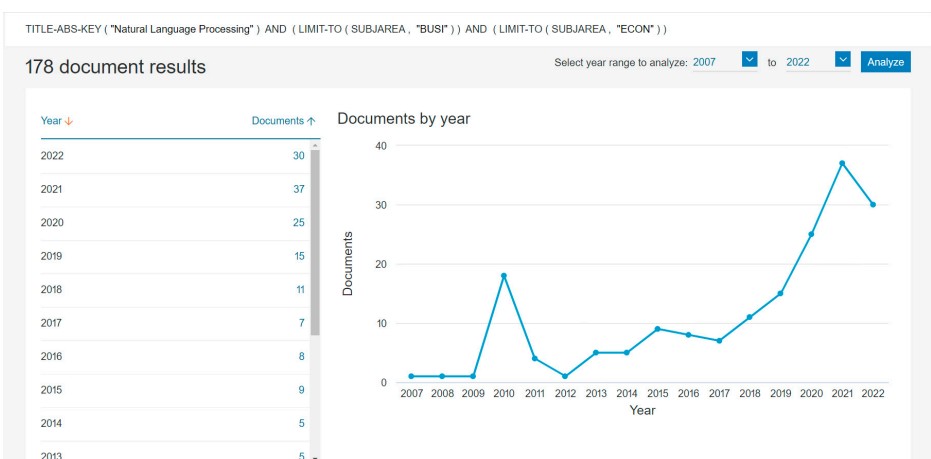

**Figure 6.** NLP literature studies in the business, economics, and finance areas.

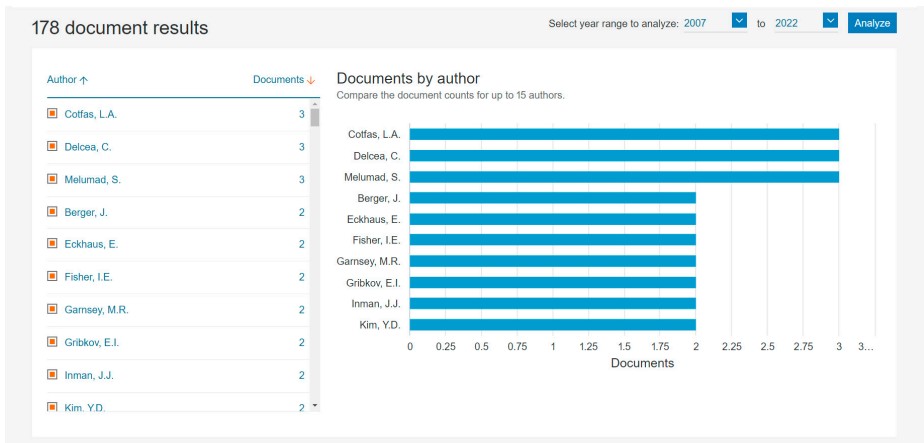

**Figure 7.** Main contributors to the NLP literature.

## 3. Methodology

The proposed model in this research is based on Vector Space Model (VSM) for document matching and similarity measures. Product proposals became a query, and policy documents by the central bank would be a corpus or database for document matching. Both the query and corpus went through a preprocessing stage prior to similarity analysis. One set of queries with two sets of corpora is tested in this research to compare similarity values. For document similarity checking, this research adopts the Levenshtein Distance

method in Python programming language. It is a technique of finding strings that match with a given string partially and not exactly. When a user misspells a word or enters a word partially, fuzzy string matching helps in finding the right word. According to [13], among the advantages of this method are it may improve data quality and accuracy and is used for fraud detection within an organization. The algorithm behind fuzzy string matching does not simply look at the equivalency of two strings but rather quantifies how close two strings are to one another. This is usually performed using a distance metric known as 'edit distance.' This determines the closeness of two strings by identifying the minimum alterations needed to convert one string into another. Figure 8 below presents the proposed algorithm for Shariah document screening.

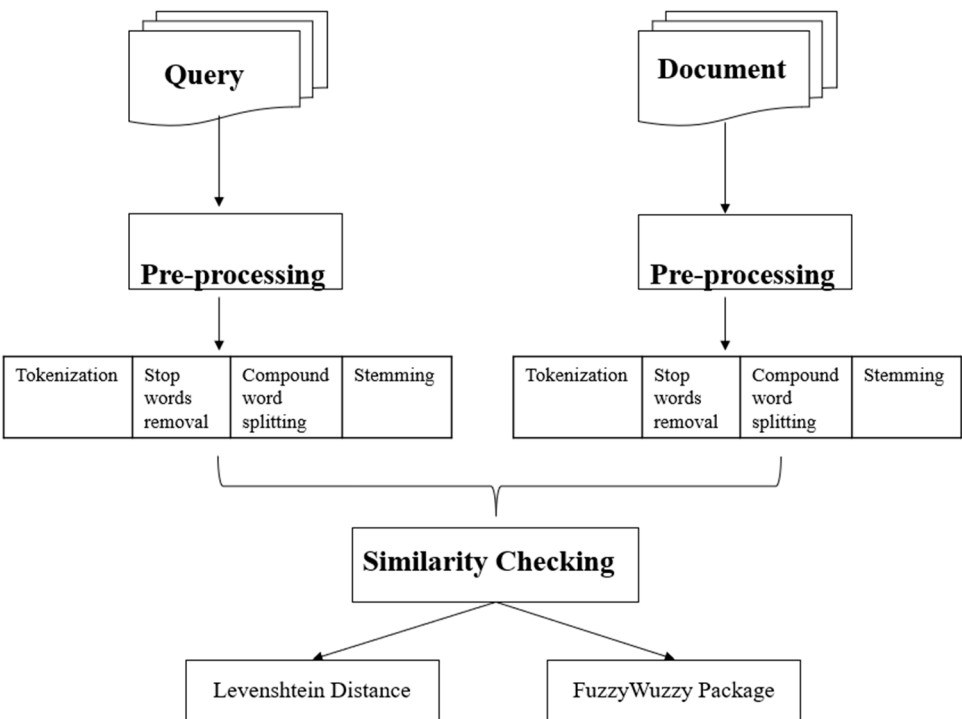

**Figure 8.** Proposed algorithm for Shariah document screening.

*3.1. Data Preprocessing*

Two sets of documents (product proposal and Central Bank of Malaysia (CNM) guidelines) were selected in this research and went through a preprocessing stage to avoid any irrelevant information or accuracy of the results later. Steps in data preprocessing include tokenization, stop-word removal, and stemming [14]. The documents first went through tokenization which the whole documents were transferred into words using white spaces. It was followed by punctuation and stop word removal such as commas, semicolons, 'and', 'is', 'or', and others. Next, any compound words were split, and stemming was accomplished using Porter stemming program. In this stage, words were converted into their stems, such as 'structuring' or 'structures' into 'structure', to determine domain vocabularies and to reduce redundancy, as most of the time, the word stem and their derived words mean the same. Once this preprocessing stage was completed, the similarity analysis was performed.

This is an important and essential step prior to the next level of model development in NLP. A set of text corpus (data) collected from one or many sources may have inconsistencies and ambiguity that requires preprocessing to clean it up. If text preprocessing is not performed properly, it may affect the output of the NLP model later. Using the NLTK library, text preprocessing procedures were conducted, which included lowercasing, removing extra whitespaces, punctuations, stopwords, tokenization, spelling correction, stemming, and lemmatization.

### 3.2. Measure of Similarity

Once the queries and corpus have been screened and are ready for further analysis, the vital process is checking for document similarities. Similarity analysis is a necessary stage in most Information Retrieval (IR) and Natural Language Processing (NLP) tasks, including document clustering, plagiarism detection, text categorization [15], and document screening. The success of IR models mostly depended on their similarity measures [16]. There were various measures of similarity, as discussed in the past literature, whereby the differences among the measures were on their functionality; a similarity measure effective in addressing one measurement problem may not be effective in another [17]. This research adopted Levenshtein Distance for similarity checking.

Levenshtein Distance

Levenshtein Distance, also known as Minimum Edit distance, is a popular method used to measure the distance between two strings. It is computed by counting the number of edits required to transform one string into another. The edits could be either the addition of a new letter, removal of a letter, and substitution. For example, the Levenshtein distance between "house" and "mouse" is 1 as only 1 edit is required to change 'h' into 'm'. There could be multiple ways of transitioning from one word to another, but Levenshtein distance chooses the smallest possible path. The more similarity between two strings, the less distance between them, and vice versa. This method is commonly used in autocompletion or autocorrection text applications such as Google search or online dictionaries. Four processes are involved in generating distance values, including creating the distances matrix, initializing the distances matrix, printing the distances matrix, and finally calculating distances between all prefixes [18].

Accordingly, this method is introduced by Vladimir Levenshtein in the year 1965. This is a mathematical formula created to measure the similarity of distance. The Levenshtein distance between two strings $a$, $b$ (of length $|a|$ and $|b|$, respectively) is given by lev$a$, $b$ ($|a|$,$|b|$) where: $1(ai \neq bi)$ is the indicator function equal to 0 when $ai \neq bi$ and equal to 1 otherwise, and lev $a$, $b$ $(i, j)$ is the distance between the first $i$ characters of $a$ and the first $j$ characters of $b$ [19]. Please refer to Equation (1).

Note that the first element in the minimum corresponds to deletion (from $a$ to $b$), the second to insertion, and the third to match or mismatch, depending on whether the respective symbols are the same.

$$lev_{a,b}(i,j) = \begin{cases} max(i,j), \; if \; min(i,j) = 0 \\ min \begin{cases} lev_{a,b}(i-1,j)+1 \\ lev_{a,b}(i, \, j-1)+1 \\ lev_{a,b}(i-1,j-1)+1_{(a_i \neq b_j)} \end{cases} , \; otherwise \end{cases} \tag{1}$$

$a$ = string #1
$b$ = string #2
$i$ = the terminal character position of string #1
$j$ = the terminal character position of string #2
The conditional $(a_i \neq b_j)$
$a_i$ refers to the character of string $a$ at position $i$
$b_j$ refers to the character of string $b$ at position $j$
Equation (1): Levenshtein Mathematical Formula.
Below is a sample of python codes conducted in this research to measure the Levenshtein Distance.

```
min_distance = 1
max_ratio = 0
max_ratio_label = 0
max_ratio_label_content = ""
for nums in StrOptions.keys():
```

```
Distance = lev.distance(String1.lower(),StrOptions[nums].lower())
Ratio = lev.ratio(String1.lower(),StrOptions[nums].lower())
print("Distance:", Distance, "Ratio:", Ratio, "", f'"{nums}"', StrOptions[nums])
if max_ratio < Ratio:
max_ratio = Ratio
max_ratio_label = nums
max_ratio_label_content = StrOptions[nums]
print('\n')
print("The least distance is:", min_distance, "The greatest ratio is:", max_ratio, "\nTawarruq
", max_ratio_label, max_ratio_label_content)
```

Given the algorithm written in Python, this research tests two different sets of text data where the first one is a sample of the Tawarruq product proposal (query) with the Tawarruq policy document (corpus), and the second set is the same Tawarruq proposal (query) with Shariah Governance Policy Document (SGPD) as corpus. The second text set is intentionally conducted to test the validity of the Levenshtein Distance algorithm in checking the similarity of two unrelated texts/documents. Figures 9 and 10 below present the results of Levenshtein Distance for two sets of text.

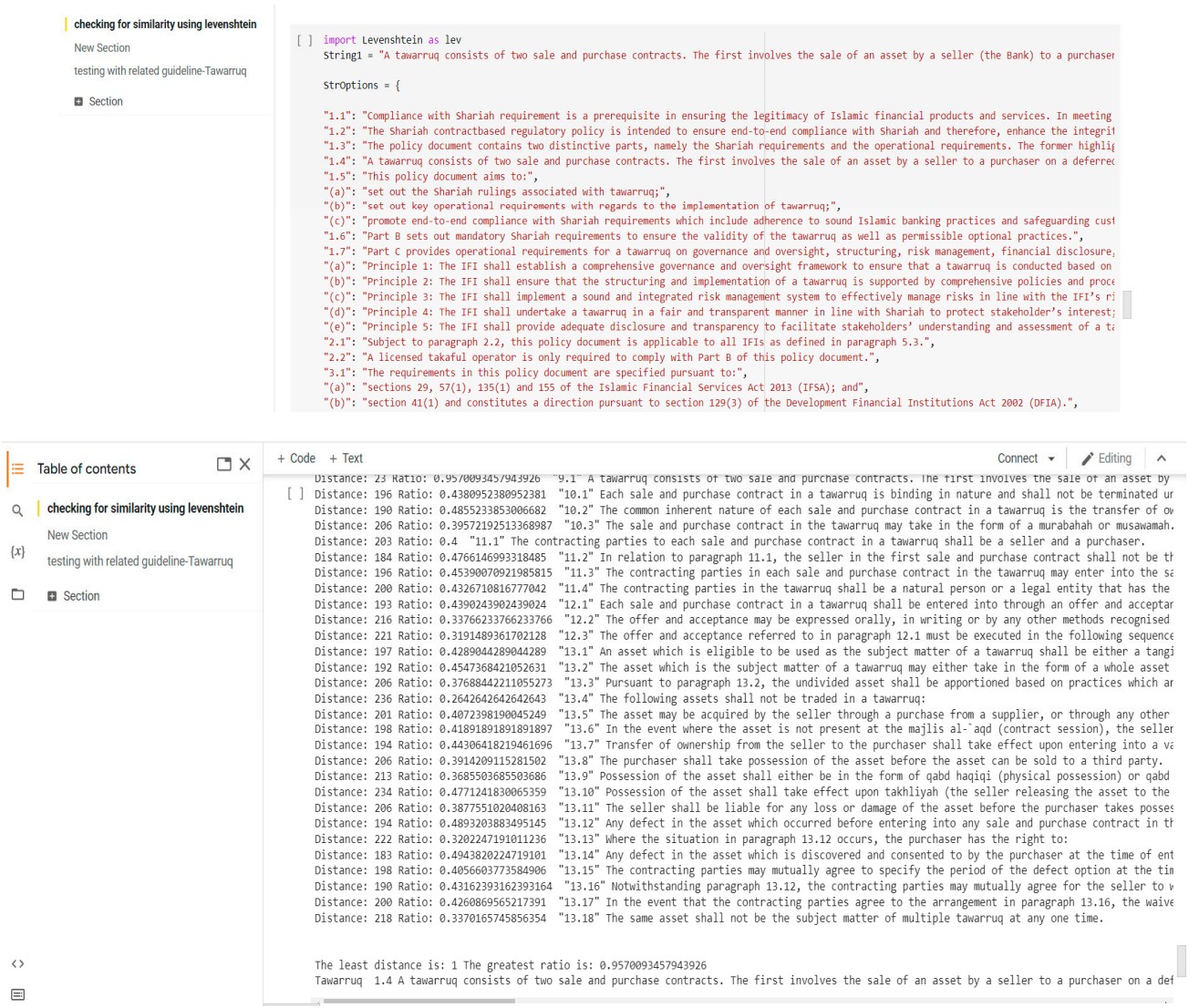

**Figure 9.** Results of Leveshtein Distance for related texts.

Based on the above similarity results, for the first set of related documents, the value of similarity is 0.957, which is near 1.0 and can be interpreted as having minimum distance or high similarity level. Meanwhile, for the second set of unrelated texts, the value of similarity is 0.451, which is far from 1.0 and is an indicator of a low level of similarity or an unrelated text document.

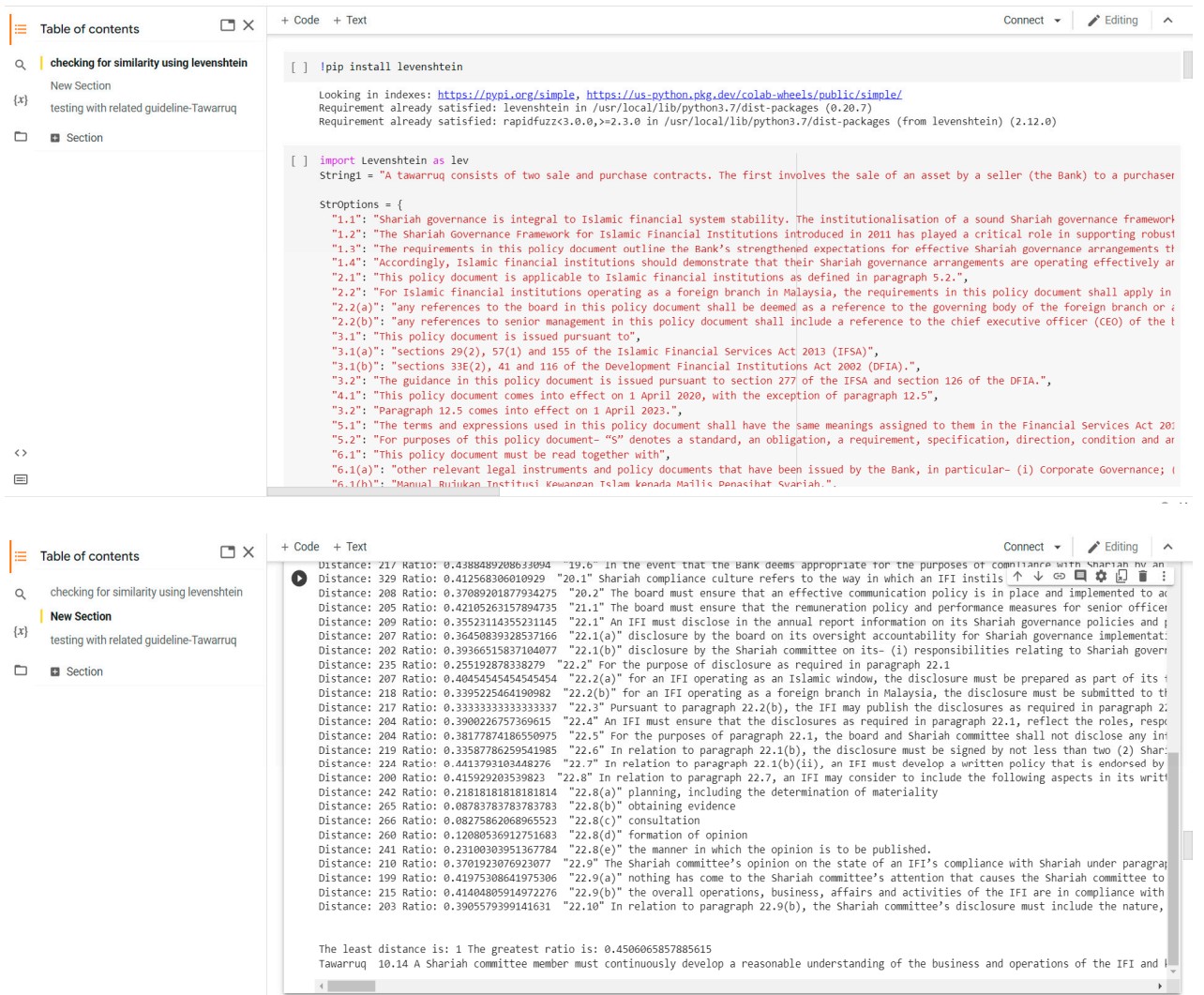

**Figure 10.** Results of Leveshtein Distance for unrelated texts.

## 4. Serverless Architecture for Shariah Document Screening Prototype

The Shariah document screening prototype is developed using a serverless architecture. A serverless architecture enables the development and operation of applications and services without the need to manage infrastructure. The application still runs on servers, but the cloud provider handles all server management. Developers are no longer required to spend time provisioning, scaling, and maintaining servers. The developers can concentrate on the core product rather than managing and operating servers or runtimes in the cloud or on-premises. This reduced overhead and allowed developers to reclaim time and resources that could be spent creating top-notch products.

### 4.1. Serverless Design Principles

Serverless architecture applications share the following design principles:
- Simplicity and speed. Concise functions should be written that are intended to carry out transactional operations or finish computing tasks applied to one or more en-

tities. These transactions must be completed quickly because they have time and capacity restrictions.

- Hardware is agnostic. It is essential, when developing serverless applications, to remove any dependencies that are hardware related. This is because the resources are only provisioned for the duration of the function's runtime.
- Optimized for concurrency. Functions should be designed considering the concurrency requests limitation of the serverless architecture. For serverless applications, optimizing for total requests may not be the best design strategy because the total request count may peak less frequently than concurrent request capacity limits.
- Temporary storage. When a function is invoked, the underlying resources are provisioned or accessed for a limited period. The state of the environment, including storage capacity, may change during the execution of a function; therefore, it may be preferable to use persistence to satisfy durable storage requirements.
- Redundancy. Failure must be handled properly by design. A single failure can propagate to subsequent requests and impede the application's operational workflow.

### 4.2. Advantages of Serverless Architecture

Serverless architecture provides many advantages because of the design principles. The advantages of serverless architecture include the following:

Deploy and run. The cloud vendor is responsible for managing infrastructure resources. Critical areas in software development can be focused on by the internal IT team. This optimizes resource usage and delivery time.

Optimized usage-based billing. The pay-as-you-go model reaches out to a bigger spectrum of industry players, including small and midsize companies.

Fault-tolerance. Hardware failures have minimal impact on the software development lifecycle due to the serverless application being logically decoupled from the underlying infrastructure.

Built-in integrations. The cloud vendors offer specialized services, including integrations and configuration work allowing software companies to focus on building high-quality applications.

Low operational overhead. Cloud vendors manage infrastructure and tasks related to operations management. The overall process of the software development lifecycle is shortened. The development team can release applications, analyze user feedback, and iterate improvements faster.

### 4.3. Disadvantages of Serverless Architecture

Security. A significant portion of data will be given to another business, which may or may not protect it. Security and apprehension about the unknown are the main reasons given by the 60% of businesses that do not use serverless systems for their operations.

Privacy. The resources are shared in cloud environments where other resources may also reside.

Complexity. When something is not working as it should, it might be difficult to pinpoint the issue. The various components involved may require significant time to troubleshoot.

Contracts. Due to the nature of the services, vendors require customers to sign lengthy contracts. These contracts are complex and may have many loopholes if not properly vetted.

### 5. Prototype Components

In this research study, the prototype is separated into two main modules. The back-end similarity analytical service is configured as a back-end service that could be scaled up automatically depending on the requests. The other module is the user interface module developed using ReactJs. Overall, the front-end is developed using ReactJS with mongodb, while the back end is based on Python language. Figure 11 below presents a sample of the Shariah document screening prototype developed in this research. Guidelines from

relevant authorities can be uploaded to the proposed back-end service component. The back-end service will be initialized with the target clauses for similarity analysis.

The current research is limited to two guideline documents. This can be extended to multiple documents from relevant authorities.

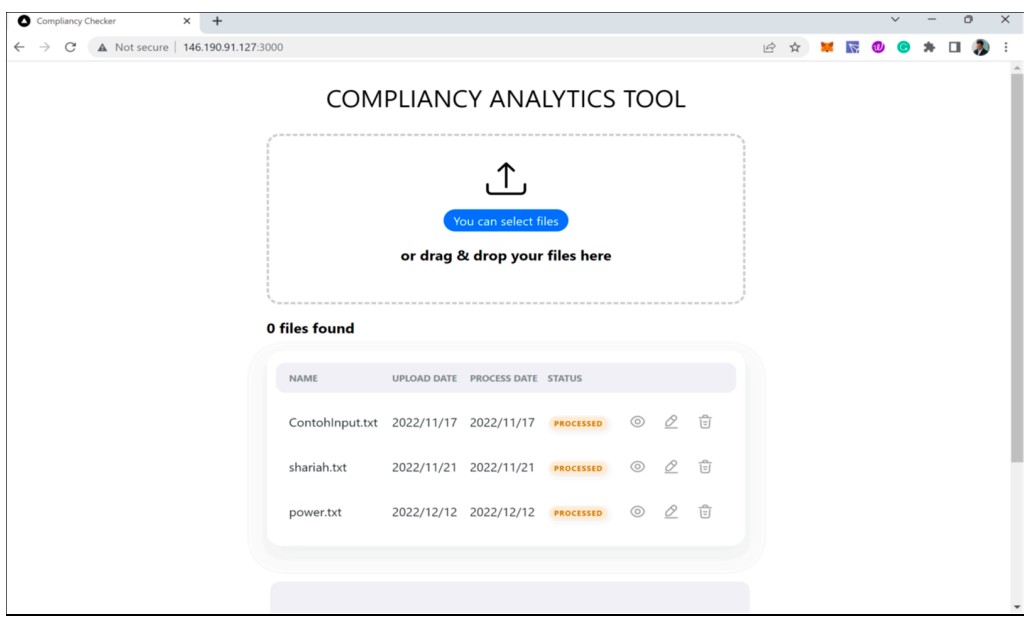

**Figure 11.** User Interface of Shariah Screening Prototype.

*Process Flow of Shariah Document Screening Prototype*

The prototype can be used by uploading a single file with the text to be analyzed for compliance against guidelines produced by the authorized agency. Once uploaded, the file will have the status "New" by default. The analysis can be started by clicking the Process button, in which the content will be extracted and passed to the back-end function for processing. The processing is performed using the Levenshtein algorithm to check for text similarity between the supplied text and the stored guidelines. Due to the nature of serverless architecture, it is possible to provide a back-end service for similarity checking using other algorithms in the future. The same front-end can be used to connect to the other service using different algorithms. It is also possible to use several algorithms and enable to assess and compare results from various algorithms used. Please refer to Figure 12.

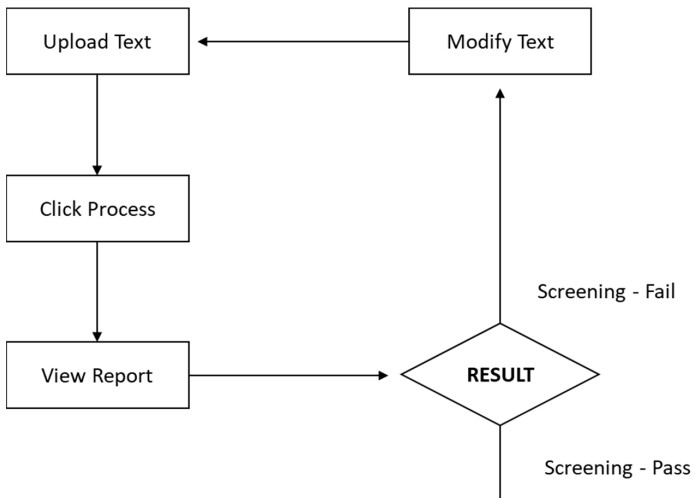

**Figure 12.** Shariah Document Screening Prototype Process Flow.

Once completed, the status of the file will be changed to Processed. The respective report in PDF format can be downloaded by clicking the View Report button. Figure 13 below is excerpted from a sample report:

**Figure 13.** Snapshot of Generated Report.

### 6. Conclusions and Discussion

The Islamic banking industry, similar to other industries, is, at present, experiencing a shift of technological advancement. On top of all-digital transactions, as financial services providers, the banks are expected to upgrade their services to customers when needed continuously. Hence, to grow with the technology and to remain compliant with industry regulations, Islamic banks must have a proper IT infrastructure and processes in place. Accordingly, applying a serverless architecture makes sense for Islamic banks as they are currently working on enhancing customer services as the demands keep growing. The adoption of serverless architecture in banking and financial services would be an added advantage to the institutions in controlling operational costs, enhancing the ecosystem, and adhering to regulatory standards, which require automation and secure distributed systems.

As the Islamic banking industry has received a special focus from the regulator mainly on the Shariah compliancy of the business, a smart, more secure, and distributed system based on a Shariah ledger of transactions must be in place. A Shariah-based automated system is needed to reduce human error and mitigate the possibility of fraud. In addition, a transparent system supported by open sources would assist the Islamic banking ecosystem. Hence, serverless architecture, which is built on a cloud computing mechanism in the context of this research study, will act as a deployment platform for the document screening process. It also mitigates the need for vast architecture knowledge and responsibilities such as management, provisioning, and maintenance. These will lower the cost of IT infrastructure for the banking ecosystem.

The Shariah document screening proposed in this research is tested using a Python programming language based on Levenshtein Distance similarity analysis. According to [13], among the advantages of this method are it may improve data quality and accuracy and is used for fraud detection within an organization. Recent research studies that have utilized this method include [20], where his research has estimated the consistency,

proficiency, and accuracy of fuzzy string matching as automated metrics of participants'
accuracy in speech intelligibility. This method is also useful for recording linkages [21],
spelling inspectors, spam detectors [22], and speech detection [23]. However, it has yet
to be explored in the banking industry, especially for document assessment. Levenshtein
Distance is basically quantifying the match between a given string and a target string based
on the number of shared characters. It is useful to quantify the accuracy of a particular
document given the standard document or guideline by authority.

The final step is developing an interface or prototype of this Shariah document screen-
ing platform, which is performed by adopting ReactJS with mongodb. React is undoubtedly
becoming the best tool to develop front-end applications in the financial industry nowa-
days, where skills in this area are highly demanded by the industry [24]. There has been
a lot of research conducted by adopting this method in various areas, including research
conducted by [25] for automated text translation, [26] for blockchain platform development,
and [27] to develop an electronic prescription system using NLP and blockchain technology.
Overall, the proposed prototype is the first attempt in the Islamic banking literature as well
as in the industry to focus on the Shariah document screening process. It has the potential
to be commercialized in the future.

## 7. Limitations and Suggestions for Future Research

A few limitations of using serverless architecture for the proposed prototype include
rising costs recently due to high demand. Further, testing and debugging might also
become a limitation in a serverless environment due to a lack of back-end process visibility.
In terms of security, because the involved data are generally from open-source such as the
central bank website and industry standards that are publicly accessible, there would be
not many issues with security, except at the user side when they enter their query. In this
context, developers need to be sure the data shared by the user/customer is kept secure.
Future research may extend the proposed prototype to include sources of the database from
various authorities (local and international levels). It is also suggested to investigate the
viability of the prototype among Islamic banking practitioners and to attain more valuable
input from them for enhancement. Future researchers may also apply the algorithms built
in this research to other areas and contexts.

**Author Contributions:** Conceptualization, M.C.M.S. and R.M.N.; Methodology, M.C.M.S., F.Y., M.A.
Software, M.C.M.S., F.Y.; Validation, R.M.N., F.Y., M.A. Formal analysis, M.C.M.S. Writing—review
and editing, M.C.M.S., R.M.N., F.Y., M.A. Project administration, M.C.M.S., R.M.N. All authors have
read and agreed to the published version of the manuscript.

**Funding:** The APC and prototype development were funded by research funds from Rabdan Academy.

**Data Availability Statement:** Part of data has been presented in main text which is dummy product
proposal and another part of data (policy document by central bank) can can be downloaded from
https://www.bnm.gov.my/banking-islamic-banking, accessed on 13 February 2023.

**Acknowledgments:** This work was supported by Research Fund from Rabdan Academy.

**Conflicts of Interest:** The authors declare no conflict of interest.

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
