# Peer review of "Constructing a Shariah Document Screening Prototype Based on Serverless Architecture"

_computers, doi:10.3390/computers12030050_

Round 1

Reviewer 1 Report

The paper aims to develop a prototype of Sharia Document Screening Based on Serverless Architecture which is helpful for Islamic bank practice, especially in fulfilling sharia principles. Several inputs so that the developed prototype can be better at selecting sharia documents:

         In this background, it is necessary to reveal what documents have been selected so far in Islamic financial institutions. Developing prototypes of unique papers that can fulfill sharia principles is essential for Islamic banks.

         In the past literature, it is necessary to disclose the screening of documents that have been applied so far and what the weaknesses are, especially for sharia documents.

         It has yet to be disclosed how the Serverless Architecture for this Document Screening Prototype selects documents that comply with sharia principles.

         It is necessary to test some instances of Islamic banks from the developed Sharia Document Screening Prototype Based on the Serverless Architecture.

         Serverless Architecture for Sharia Document Screening Prototypes must be added with weaknesses (risks) and prototype security.

Reviewer 2 Report

Abstract: Please focus the abstract on your study and your results.

The authors should specify more details regarding the Experiment for the proposed algorithm.
The authors should provide more details regarding the analysis of the results.

I suggest a significant rewrite of the introduction. It should provide an overview of the importance of the main contribution of the proposed algorithm.
The advantage and disadvantages of the work are suggested to be highlighted in comparison with extant studies or methods.
How to initialize the agents in the proposed Algorithm?

Some additional experiments are required:
a. - Scalability
b. - Runtime
c. - Memory
d. - Sensitivity analysis

It is necessary to discuss the complexity of the proposed Algorithm.

Read and cite these references.
E.-S. M. El-kenawy, H. F. Abutarboush, A. W. Mohamed and A. Ibrahim, “Advance artificial intelligence technique for designing double T-shaped monopole antenna,” Computers Materials & Continua, vol. 69, no. 3, pp. 2983-2995, 2021.

Some syntax errors or improper expressions exist in the manuscript.

More up-to-date studies are suggested to be cited.

Author Response

Response which is not mentioned in the attachment;

  • Some additional experiments are required: 
    • - Scalability
    • - Runtime
    • - Memory
    • - Sensitivity analysis 

Response->These additional experiments may be considered for future research.

  • It is necessary to discuss the complexity of the proposed Algorithm. 

Response ->The proposed algorithm is based on a well-studied similarity findings between inputs. The paper established acceptable result of using the proposed algorithm with manual process performed by experts. Other algorithm could be added for future research. One potential algorithm for future research include Siamese Network algorithm in which can be used to find similarities.

Reviewer 3 Report

I could not find any major corrections for the main body of the paper.

(i.) However, there are many editorial corrections that need to be made for several of the References that are currently have inconsistent formats, such as listed below:

(1.) References [2], [8], [9], [16], and [17] use full first names when all others use only initials that is usually the standard format for other journals.

(2.) References [3], [4], [6], [7], [11], [18], [19] , [21] and [22] have titles that do not use capitalization for the words of the titles beyond the first word of title, when References [1], [2], [5], [8], [10], [12], [13], [16], [17] do use capitalization for each of the major words of titles.

(3.) Reference [9] has no title.

(4.) Reference [15] has no text other a URL. This is an incomplete format and should at least include a title of what the URL represents or contains as in the following reference [16].

(ii.) NLP is used many times in this paper but is only spelled out once in line 40 and should be spelled out again because it used so many times.

(iii.) Figure 10 on page 11 has two words of "Fail" and "Pass" that are not completely contained within the boundaries of this Figure.

(iv.) Figure 11 on page 11 has no blank line preceding "Clause: 1.4=>" when all of the other Clauses in this Figure do.

(iv.) Figure 1 on page 3 has title that appears on the next page 4, and should appear directly below Figure 1 on same page 3.

(v.) "Figure 3" in title on line 140 of page 4 is not in bold as in titles of all other Figures in this paper.

(vi.) Title of Section 4 is missing space after "4." and before "S" in "Serverless".

(vii.) Font of title for Figure 10 in line 312 on page 11 is smaller font size than titles of all other figures such as that of Figure 11 in line 317 of same page 11.

Author Response

Please see the attachment. TQ

Round 2

Reviewer 1 Report

Thank you, some of the comments given have been corrected as expected in the revised manuscript. Again, congratulations; your article deserves to be published.

Author Response

TQ very much for your recommendation.

Reviewer 2 Report

Some syntax errors or improper expressions exist in the manuscript.
More up-to-date studies are suggested to be cited.

Author Response

Dear Reviewer,

Thank you for all comments.

Please see the attachment for second round responses.

TQ
